# Integrative Metabolomic and Transcriptomic Analysis Elucidates That the Mechanism of Phytohormones Regulates Floral Bud Development in Alfalfa

**DOI:** 10.3390/plants13081078

**Published:** 2024-04-11

**Authors:** Xiuzheng Huang, Lei Liu, Xiaojing Qiang, Yuanfa Meng, Zhiyong Li, Fan Huang

**Affiliations:** Institute of Grassland Research, Chinese Academy of Agricultural Sciences, Hohhot 100081, China; 19917621903@163.com (X.H.); nymyf1990@163.com (Y.M.); zhiyongli1216@126.com (Z.L.); huangfan@caas.cn (F.H.)

**Keywords:** *Medicago sativa*, transcriptome and metabolome, floral bud, phytohormones

## Abstract

Floral bud growth influences seed yield and quality; however, the molecular mechanism underlying the development of floral buds in alfalfa (*Medicago sativa*) is still unclear. Here, we comprehensively analyzed the transcriptome and targeted metabolome across the early, mid, and late bud developmental stages (D1, D2, and D3) in alfalfa. The metabolomic results revealed that gibberellin (GA), auxin (IAA), cytokinin (CK), and jasmonic acid (JA) might play an essential role in the developmental stages of floral bud in alfalfa. Moreover, we identified some key genes associated with GA, IAA, CK, and JA biosynthesis, including *CPS*, *KS*, *GA20ox*, *GA3ox*, *GA2ox*, *YUCCA6*, *amid*, *ALDH*, *IPT*, *CYP735A*, *LOX*, *AOC*, *OPR*, *MFP2*, and *JMT*. Additionally, many candidate genes were detected in the GA, IAA, CK, and JA signaling pathways, including *GID1*, *DELLA*, *TF*, *AUX1*, *AUX*/*IAA*, *ARF*, *GH3*, *SAUR*, *AHP*, *B-ARR*, *A-ARR*, *JAR1*, *JAZ*, and *MYC2*. Furthermore, some TFs related to flower growth were screened in three groups, such as *AP2*/*ERF*-*ERF*, *MYB*, *MADS*-*M*-*type*, *bHLH*, *NAC*, *WRKY*, *HSF,* and *LFY*. The findings of this study revealed the potential mechanism of floral bud differentiation and development in alfalfa and established a theoretical foundation for improving the seed yield of alfalfa.

## 1. Introduction

The development of floral buds encompasses the initiation of plant reproductive growth, and they directly affect the yield and quality of seeds [1,2]. Researchers usually classify the stages of flower bud development by observing the morphological changes during flower bud growth [3,4]. Additionally, studying the dynamic changes in the physiological indexes and gene expression in each stage of flower bud growth can provide a theoretical basis for uncovering the potential mechanism of flower organ formation, improving the seed yield [5,6].

Alfalfa (*Medicago sativa*) is a perennial leguminous plant cultivated worldwide [7]. It is the most important forage in stockbreeding, and it has good quality, high yield, and strong resistance to stress [8,9]. However, low seed yields have limited the long-term development of the alfalfa industry. Alfalfa inflorescences are typical racemes with small flowers growing on the peduncle. Previous studies have revealed that the floret count determines the number of seeds, and the number of seeds per inflorescence is positively correlated with the seed yield [10,11]. Moreover, the floral bud growth affects the quantity and quality of florets and seeds in plants [12]. However, a few investigations have been performed on the inherent mechanisms of floral bud development in alfalfa.

The external environment and internal factors commonly influence the development of floral buds [13,14,15]. Endogenous phytohormones play an important role in regulating flower bud growth [16,17]. Various endogenous hormones, such as auxins (IAA), cytokinin (CK), abscisic acid (ABA), jasmonate (Ja), gibberellin (GA), and ethylene (ETH), coordinate with each other, thereby regulating flower bud development by jointly controlling the metabolism of plants. Yan et al. (2019) found that a higher IAA content was beneficial to the differentiation of licorice flower buds, eventually improving the seed yield of licorice [18]. Before floral bud growth, increasing the levels of ABA and GA3 in stem apical meristems may promote flower induction and floral bud growth [19]. Fang et al. (2018) explored the effect of growth regulators on cotton floral buds and concluded that the ratio of trans-Zeatin-riboside (ZR)/IAA and GA3/IAA was significantly elevated in the seeds after pretreatment with GA3 and N6-benzyladenine (6-BA), resulting in an increase in the number of floral buds [20]. During the process of grape floral bud differentiation, GAs inhibited the primordium differentiation of grape inflorescences, and CTK could promote grape flowering [21,22]. Understanding the dynamic change in endogenous phytohormones in the developmental stages of floral buds can establish a theoretical foundation for conducting cultivation management and improving the seed yield of alfalfa.

In recent years, with the development of high-throughput sequencing, omics technology has become an important method to explore the potential mechanism of floral bud growth. For example, Qu et al. demonstrated that GA_3_ played a crucial role in regulating the floral bud development of *Cyclocarya paliurus* based on transcriptome analysis [23]. Moreover, Xie et al. analyzed the gene expression and identified candidate genes associated with female and male floral bud development in *Carya illinoensis* by using RNA sequencing [24]. In this work, we investigated the gene expression and phytohormone accumulation in three developmental stages of floral bud development in alfalfa, which provided a theoretical basis for molecular breeding.

## 2. Results

### 2.1. Transcriptome Analysis

By observing the morphological changes in floral bud development of alfalfa, floral bud growth is divided into three developmental stages (D1, D2, and D3) (Figure 1A,B). At the early bud stage (D1), the floral buds are detectable as small swellings, surrounded by the leaf primordium. At the mid bud stage (D2), the floral bud is larger and easier to detect. Small swellings differentiate into multiple florets along with floral buds enlarging, and many white trichomes grow on the floret primordium. At the late bud stage (D3), each floret primordium becomes mature along with the bud expanding and lengthening rapidly. After the late bud stage, the florets will gradually bloom. These three stages are the most typical stages in the developmental process of floral buds.

Floral buds at the three stages (D1, D2, and D3) were utilized in transcriptome sequencing to elucidate the potential mechanism of floral bud development in alfalfa. As a result, a total of 59.21 Gb clean data were identified, and the clean data of each sample reached 5 Gb. Moreover, the Q30 base percentage was above 94%, and more than 80% of the clean reads were mapped to the *Medicago sativa* reference genome (Appendix A). Principal component analysis (PCA) suggested that the three samples were obviously separated (Figure 2A). These results indicated that the transcriptome datasets were reliable and accurate for further investigation. Furthermore, a total of 1736 up-regulated and 407 down-regulated genes, 3131 up-regulated and 1604 down-regulated genes, and 1465 up-regulated and 1373 down-regulated genes were identified in the D2 vs. D1, D3 vs. D1, and D3 vs. D2 comparisons, respectively (Figure 2B). In addition, nine genes were selected to identify the reliability of the transcriptome datasets. The results showed that the FPKM value of most genes had a similar tendency to the relative expression levels in the three groups, revealing that the transcriptome data in this work can be trusted for further investigation (Appendix A).

### 2.2. GO and KEGG Enrichment Analysis

Subsequently, we conducted a GO and KEGG enrichment analysis of differentially expressed genes (DEGs) in the D2 vs. D1 and D3 vs. D2 comparisons. In the D2 vs. D1 comparison, the GO enrichment results revealed that most DEGs were mainly enriched in biological processes, cellular components, and molecular functions, including the anatomical structure formation involved in morphogenesis (GO:0048646), secondary metabolite biosynthetic process (GO:0044550), plant-type cell wall organization or biogenesis (GO:0071669), integral component of plasma membrane (GO:0005887), glucosyltransferase activity (GO:0046527), inorganic anion transmembrane transporter activity (GO:0015103), and some flower-development-related terms (Appendix A). In the D3 vs. D2 comparison, most DEGs were mainly enriched in biological processes and molecular functions, including anatomical structure formation involved in morphogenesis (GO:0048646), secondary metabolite biosynthetic process (GO:0044550), cell wall biogenesis (GO:0042546), glucosyltransferase activity (GO:0046527), UDP-glucosyltransferase activity (GO:0035251), dioxygenase activity (GO:0045543), and some flower-development-related terms (Appendix A).

In addition, KEGG enrichment showed that many genes were enriched in metabolic pathways, biosynthesis of secondary metabolites, phenylpropanoid biosynthesis, protein processing in the endoplasmic reticulum, and flavonoid biosynthesis in the D2 vs. D1 comparison (Figure 3A). In the D3 vs. D2 comparison, most DEGs were enriched in metabolic pathways, biosynthesis of secondary metabolites, plant hormone signal transduction, and phenylpropanoid biosynthesis (Figure 3B). These results suggested that these pathways might play a critical role in floral bud development.

### 2.3. Metabolomic Analysis

From the KEGG enrichment results, we discovered that many DEGs were enriched in phytohormone-biosynthesis-related pathways, such as tryptophan metabolism (ko00380), carotenoid biosynthesis (ko00906), diterpenoid biosynthesis (ko00904), alpha-Linolenic acid metabolism (ko00592), zeatin biosynthesis (ko00908), and cysteine and methionine metabolism (ko00270) (Figure 3A,B). These results indicated that phytohormones are closely connected to the floral bud development.

To detect the content of phytohormones in floral buds at the three stages (D1, D2, and D3), a targeted phytohormone metabolome analysis was conducted. The PCA results showed that the three groups had a good separation, suggesting that the experimental results were reliable for further study (Figure 4A). Subsequently, a total of 17, 31, and 25 differentially accumulated metabolites (DAMs) were identified in the D2 vs. D1, D3 vs. D1, and D3 vs. D2 comparisons, respectively (Figure 4B). Moreover, seven categories of plant hormones were differentially accumulated in the three groups, including cytokinin (CK), auxin (IAA), jasmonic acid (JA), salicylic acid (SA), gibberellin (GA), abscisic acid (ABA), and ethylene (ETH) (Figure 4C). These results suggested that these phytohormones might participate in the floral bud development.

### 2.4. Quantitative Analysis of Differentially Accumulated Phytohormones

Furthermore, we performed a quantitative analysis of DAMs in the D2 vs. D1 and D3 vs. D2 comparisons to investigate the dynamic changes in seven phytohormones in the developmental process of floral bud (Table 1 and Table 2). For GA, GA8 (Log_2_ fold-change value = Inf; Log_2_ fold-change value = 1.07) was up-accumulated in the D2 vs. D1 and D3 vs. D2 comparisons, and the content of GA7 (Log_2_ fold-change value = 1.13) was higher in the D3 group compared to the D2 group (Table 1 and Table 2).

For auxin, one up-accumulated (IPA, Log_2_ fold-change value = Inf) and two down-accumulated (IAA, Log_2_ fold-change value = −Inf; TRA, Log_2_ fold-change value = −Inf) auxins were detected in the D2 vs. D1 comparison. Additionally, IAA (Log_2_ fold-change value = Inf), IAA-Glu (Log_2_ fold-change value = 1.98), TRA (Log_2_ fold-change value = Inf), IAA-Asp (Log_2_ fold-change value = 1.05), and IA (Log_2_ fold-change value = 1.33) were up-accumulated in the D3 vs. D2 comparisons, and IPA did not accumulate in the D3 group (Table 1 and Table 2).

Furthermore, we found that nine up-accumulated and two down-accumulated CKs, and ten up-accumulated and two down-accumulated CKs, were detected in the D2 vs. D1 and D3 vs. D2 comparisons, respectively. Notably, we found that pT9G (Log_2_ fold-change value = 1.06; Log_2_ fold-change value = 1.23), iP9G (Log_2_ fold-change value = Inf; Log_2_ fold-change value = 1.20), tZ (Log_2_ fold-change value = Inf; Log_2_ fold-change value = 2.40), DHZ7G (Log_2_ fold-change value = 1.13; Log_2_ fold-change value = 2.59), tZR (Log_2_ fold-change value = Inf; Log_2_ fold-change value = 3.77), and tZRMP (Log_2_ fold-change value = Inf; Log_2_ fold-change value = 3.12) were significantly up-accumulated in the D2 vs. D1 and D3 vs. D2 comparisons (Table 1 and Table 2).

For JA, MEJA (Log_2_ fold-change value = 1.19; Log_2_ fold-change value = 1.04) was up-accumulated in the D2 vs. D1 and D3 vs. D2 comparisons, and 12-OH-JA (Log_2_ fold-change value = Inf) maintained a high accumulation in the D3 group compared to the D2 group. For ETH, we found that ACC (Log_2_ fold-change value = Inf) was up-accumulated in the D3 vs. D2 comparison. For ABA, ABA-ald (Log_2_ fold-change value = −Inf) and ABA-GE (Log_2_ fold-change value = −Inf) were down-accumulated in the D2 vs. D1 and D3 vs. D2 comparisons, respectively. For SA, t-CA (Log_2_ fold-change value = −1.21) was down-accumulated in the D3 vs. D2 comparison (Table 1 and Table 2).

Based on the results above, we concluded that the content of one GA (GA8), six CKs (pT9G, iP9G, tZ, DHZ7G, tZR, and tZRMP), and one JA (MEJA) continuously increased with development, and some auxins were differentially accumulated in the D3 vs. D2 comparison. Therefore, we speculated that GA8, pT9G, iP9G, tZ, DHZ7G, tZR, and tZRMP, and MEJA might be involved in floral bud growth, and auxins played a vital role in the mid and late bud stages.

### 2.5. Key Genes Involved in GA, IAA, CK, and JA Biosynthesis Pathways

To screen the candidate genes involved in crucial phytohormone synthesis, the GA, IAA, CK, and JA biosynthesis pathways were examined, respectively (Figure 5). The FPKM value of DEGs in each group is shown in Appendix A. In GA biosynthesis, we identified that one *CPS* (*MsG0780036356.01*), three *GA20ox* (*MsG0180005519.01*; *MsG0180004530.01*; *novel.4751*), and five *GA2ox* (*MsG0780038278.01*; *MsG0880046136.01*; *MsG0580024092.01*; *MsG0280010379.01*; *MsG0280007031.01*) genes were differentially expressed in the D2 vs. D1 comparison, and two *KS* (*novel.7393*, *novel.6028*), two *GA20ox* (*MsG0180005519.01*; *MsG0380016852.01*), GA3ox (*MsG0280011347.01*), and five *GA2ox* (*MsG0580024092.01*; *MsG0180004772.01*; *MsG0280009804.01*; *MsG0280007350.01*; *MsG0480021427.01*) genes were differentially expressed in the D3 vs. D2 comparison.

In IAA biosynthesis, one *YUCCA6* (*MsG0380017591.01*), two *amid* (*MsG0480022990.01*; *MsG0480021433.01*), and one *ALDH* (*MsG0880045054.01*) genes were differentially expressed in the D3 vs. D2 comparison.

In the CK biosynthesis, one *IPT* (*MsG0180005958.01*) and two *CYP735A* (*novel.6718*; *MsG0780036054.01*) genes were significantly up-regulated in the D2 vs. D1 and D3 vs. D2 comparisons, respectively.

In JA biosynthesis, one *LOX* (*novel.8870*), one *AOC* (MsG0480021206.01), two *OPR* (*MsG0580024179.01*; *MsG0580024168.01*), one *MFP2* (*MsG0880047061.01*), and three *JMT* (*MsG0080049067.01*; *MsG0180000127.01*; *MsG0180000128.01*) genes were differentially expressed in the D2 vs. D1 comparison, and two *LOX* (*MsG0380015883.01*; *novel.8870*), one *AOC* (*MsG0480021206.01*), and three *OPR* (*MsG0580024168.01*; *MsG0580024178.01*; *MsG0180000008.01*) genes were differentially expressed in the D3 vs. D2 comparison. These DEGs might be involved in GA, IAA, CK, and JA accumulation in floral bud development.

### 2.6. Key Genes Involved in GA, IAA, CK, and JA Signaling Pathways

Simultaneously, some DEGs were identified in the GA, IAA, CK, and JA signaling pathways in the D2 vs. D1 and D3 vs. D2 comparisons (Figure 6). The FPKM value of DEGs in each group is shown in Appendix A. In the GA signaling pathway, four *GID1* (*MsG0180000605.01*; *MsG0780040621.01*; *MsG0680034380.01*; *MsG0780041225.01*), three *DELLA* (*MsG0580024757.01*; *novel.1151*; *MsG0280010628.01*), and two *TF* (*novel.8984*; *MsG0880045870.01*) genes were differentially expressed in the D2 vs. D1 and D3 vs. D2 comparisons.

In the auxin signaling pathway, two *AUX1* (*novel.3161*; *MsG0480021449.01*), five *AUX*/*IAA* (*MsG0880046805.01*; *MsG0280011343.01*; *MsG0180003906.01*; *MsG0180003907.01*; *MsG0880047279.01*), nine *ARF* (*MsG0580029898.01*; *novel.8654*; *MsG0580026167.01*; *MsG0280007416.01*; *MsG0580029167.01*; *novel.1363*; *MsG0580027493.01*; *MsG0580027483.01*; *MsG0380015618.01*), one *GH3* (*MsG0580024778.01*), and nine *SAUR* (*MsG0480021377.01*; *MsG0780037963.01*; *MsG0480021757.01*; *MsG0580025075.01*; *MsG0480021346.01*; *MsG0480021367.01*; *MsG0480021370.01*; *MsG0180003514.01*; *MsG0780040970.01*) genes were differentially expressed in the D2 vs. D1 and D3 vs. D2 comparisons.

In the CK signaling pathway, two *AHP* (*MsG0180004543.01*; *MsG0780041591.01*), four *B-ARR* (*MsG0480023946.01*; *MsG0480018999.01*; *MsG0880046698.01*; *MsG0280006898.01*), and five *A-ARR* (*MsG0780040418.01*; *MsG0380016366.01*; *MsG0580025893.01*; *MsG0380015850.01*; *MsG0380012160.01*) genes were differentially expressed in the D2 vs. D1 and D3 vs. D2 comparisons.

In the JA signaling pathway, one *JAR1* (*MsG0880043041.01*), one *JAZ* (*novel.7289*), and eight *WYC2* (*MsG0480022831.01*; *MsG0280011447.01*; *novel.10430*; *MsG0180003989.01*; *MsG0880042940.01*; *MsG0880042939.01*; *MsG0580025575.01*; *MsG0880042918.01*) genes were differentially expressed in the D2 vs. D1 and D3 vs. D2 comparisons.

### 2.7. Transcription Factor Analysis

In this work, we identified a total of 2837 TFs (transcription factors) in the three groups, including 190 *AP2/ERF-ERF* (6.7%), 147 *FAR1* (5.18%), 146 *MYB* (5.15%), 140 *MADS-M-type* (4.93%), 134 *bHLH* (4.72%), 120 *NAC* (4.23%), and 118 *B3* (4.16%) families (Figure 7A), suggesting these TF families might play an essential role in regulating floral bud growth. Moreover, a total of 156 and 172 differentially expressed TFs were detected in the D2 vs. D1 and D3 vs. D2 comparisons, respectively. Among these differentially expressed TFs, 17 TFs with continuously increased expression levels were screened in the three developmental stages, including 1 *AP2*/*ERF*-*ERF*, 4 *bHLH*, 1 *C2H2*, 1 *HSF*, 2 *MADS*-*MIKC*, 1 *7* 1 *TCP*, and 1 *WRKY* (Figure 7B). Simultaneously, 18 TFs with persistently decreased expression levels were identified in the three developmental stages, including 3 *B3*, 1 *bHLH*, 2 *bZIP*, 1 *GARP*-*G2*-like, 1 *GRAS*, 2 *HB*-*HD*-*ZIP*, 1 *LFY*,1 *MADS*-*MIKC*, 1 *MYB*, 1 *MYB*-related, 2 *NAC*, 1 *PLATZ*, and 1 *SBP* (Figure 7B). The FPKM value of these differentially expressed TFs in each group is shown in Appendix A. These TFs might be closely correlated with the floral bud development.

## 3. Discussion

The development of floral buds determines the seed yield. To explore the gene expression patterns and phytohormone accumulation of the floral bud differentiation and development in alfalfa, we performed a comparative transcriptome and metabolome analysis for the three developmental stages of floral buds.

Plant hormones are the key regulatory factors during plant morphogenesis [16,17]. Spraying growth regulators on a plant is a conventional cultivation method to increase the crop yield during the flower bud developmental stage; therefore, exploring the accumulation of endogenous phytohormones in floral buds can provide a theoretical support for guiding the production and improving the seed yield. Cytokinin plays a key role in reproductive growth [25]. Gibberellin can affect flower bud differentiation by regulating the expression of downstream genes [26]. Auxin is a crucial plant hormone involved in the development of flower organs [27]. It is involved in regulating the gibberellin signaling pathway and eventually promoting flower formation [28]. Jasmonic acid can participate in floral organ development via promoting *SlMYB21* expression in tomatoes [29]. In agreement with our findings, we found that the content of one GA (GA8), six CKs (pT9G, iP9G, tZ, DHZ7G, tZR, and tZRMP), and one JA (MEJA) continuously increased in the floral bud developmental process, and auxins were significantly differentially expressed in the D2 vs. D1 and D3 vs. D2 comparisons. These results indicated that these phytohormones might play a vital role in regulating the downstream genes related to flower organ development. Notably, previous studies have shown that auxin polar transport is closely related to plant morphological construction [30,31]; therefore, identifying the accumulation in specific cells of auxin will be important for exploring the mechanism of auxins involved in floral bud development in the future.

Moreover, some candidate genes related to GA, IAA, CK, and JA biosynthesis were identified based on the integrated transcriptome and metabolome results. CPS and KS are upstream enzymes of GA biosynthesis, catalyzing geranylgeranyl diphosphate to ent-Kaurene. The inhibition of CPS and KS enzymes influenced the GA biosynthesis and limited plant organ growth [32,33]. A previous study reported that the high expression levels of the *SoGA20ox1* gene in shoot tips enhanced GA biosynthesis [34]. Additionally, overexpression of the *GA2ox* gene facilitated pollen growth in transgenic *Arabidopsis* plants [35]. In this work, we detected that the expression of one *CPS*, two *KS*, four *GA20ox1*, and five *GA2ox* genes maintained a high level in the D2 and D3 groups, suggesting that these genes might play an essential role in GA8 accumulation. Moreover, Liu et al. found *FvYUCCCA6* played a critical role in vegetative and reproductive development in woodland strawberry [36]. Consistent with our results, one *YUCCA6* gene maintained high expression levels in the D3 group compared to the D1 and D2 groups, and it might be involved in the IAA accumulation in the D3 group. IPT was the first enzyme participating in CK biosynthesis, catalyzing ATP, ADT, and AMP to iPRTP, iPRDP, and iPRMP, respectively [37,38]. Then, *CYP735* converted iP-nucleotide to tZ-nucleotide [39]. In the present work, the expression levels of one *IPT* and two *CYP735* genes were high in the D2 and D3 groups, suggesting that these genes may be closely related to CK accumulation in the floral bud developmental process. In addition, we identified that two *LOX*, one *AOC*, four *OPR*, one *MFP2*, and three *JMT* genes were highly expressed in the D2 and D3 groups. Previous studies have demonstrated that these genes have vital functions in JA biosynthesis [40,41]. An enhancement or inhibition of gene expression in these genes directly affects JA synthesis, eventually changing the plant physiological activities [42,43,44,45,46]. Therefore, these DEGs related to JA biosynthesis might play an essential role in regulating the floral bud growth.

The phytohormone signaling pathway was associated with plant organ development [47]. In accordance with our study, many DEGs were enriched in the plant hormone signal transduction pathway. The findings of a previous study revealed that *GID1* and *TF* positively promote flower bud differentiation [48]. Consistent with our results, several *GID1* and *TF* genes in the D2 or D3 group were highly expressed in the GA signaling pathway. Additionally, most studies demonstrated that the DELLA protein inhibits plant development and growth by binding to transcription factors [49,50,51]. In this work, the expression level of most *DELLA* genes was low in the D3 group, suggesting that the *DELLA* protein played a crucial role in participating in floral bud differentiation. In the auxin signaling pathway, the *AUX1*, *TIR*, *AUX*/*IAA*, *ARF*, *GH3*, and *SAUR* genes played a pivotal role in regulating plant growth [52]. In the present work, most differentially expressed *AUX1*, *AUX*/*IAA*, *ARF*, and *SAUR* genes in the auxin signaling pathway maintained high expression levels in the D2 and D3 groups compared to the D1 group, revealing that the auxin signaling pathway may participate in regulating floral bud development. *A*-*ARR* was a key gene involved in the CK signaling pathway and affected flower organ development and growth [53,54]. In this work, the expression level of four *A*-*ARR* genes was significantly higher in the D3 group than in the D1 and D2 groups, suggesting *A*-*ARR* genes might play a crucial role in regulating floral bud development in the D3 group. In the JA signaling pathway, *JAR1* can induce the JA converted to the biologically active JA-Ile by responding to environmental stress [40], and the JAZ protein family usually regulates JA responses by interacting with the *MYC* family [55]. As we are currently aware, one *JAR1*, one *JAZ,* and eight *MYC* genes were differentially expressed in the three developmental stages, and these genes might be positively involved in floral bud development.

In this work, many *AP2*/*ERF*-*ERF*, *MYB*, *MADS*-*M*-*type*, *bHLH*, *NAC*, and *WRKY* genes were identified in the three developmental stages, which was consistent with previous studies [56,57]. AP2/ERF-ERF transcription factor family proteins are mainly involved in the development of sepals and petals in reproductive organs [58]. The overexpression of *RcAP2* results in the transformation of stamens into petals, thereby increasing the number of petals in *Arabidopsis*, while silencing *RcAP2* decreases the number of petals [59]. *MYB* transcription factors play a role in the flowering time, pollen development, flower color, and sex differentiation of flower organs [60]. During the developmental stage of *Arabidopsis* reproductive organs, *AtMYB125* and *AMYB98* are involved in the development of male and female gametes, respectively [61,62]. Most studies indicated that the *MADS* family is an important factor regulating flowering time and flower organ development in plants [63,64,65]. bHLH proteins CIB1, CIB2, CIB4, and CIB5 commonly regulate flowering initiation, which facilitates *FT* transcription by binding to the *FT* promoter in plants [66]. Additionally, *AtWRKY75* may be a new member regulating flowering in the GA signaling pathway, and *WRKY71* induces early flowering by activating *FT* and *LFY* in *Arabidopsis* [67,68]. The *NAC* transcription factor family can regulate flower growth by participating in the JA and GA signaling pathways [69,70]. Simultaneously, we also discovered that the expression level of some TFs, such as *AP2*/*ERF*-*ERF*, *MYB*, *MADS*-*M*-*type*, *bHLH*, *NAC*, *WRKY*, *HSF,* and *LFY,* is continuously increased or decreased in the three developmental stages, suggesting that these TFs might play a pivotal role in regulating floral bud development.

## 4. Materials and Methods

### 4.1. Plant Materials

Alfalfa seedlings were planted at the Institute of Grassland Research, Chinese Academy of Agricultural Sciences, Hohhot (40°58′ N, 111°78′ E). The samples of the three developmental stages were collected in the early flowering period. We observed the monological traits of floral buds and identified the developmental stages of floral buds by using a dissecting microscope (Dian Ying, Shanghai, China). The floral buds of the three developmental stages were named D1, D2, and D3. Three biological replicates were obtained for the three samples, and the mass of each biological replicate was more than 3 g. All samples were stored in liquid nitrogen.

### 4.2. Transcriptome Sequencing and Data Analysis

By utilizing ethanol precipitation and CTAB-PBIOZOL, the total RNA of floral buds (D1, D2, and D3) was obtained. Total RNA was analyzed by utilizing a Qubit fluorescence quantifier and a Qsep400 high-throughput biofragment analyzer (AUTO Q BIOSCIENCES, San Diego, CA, USA). Subsequently, all cDNA libraries were sequenced on the Illumina platform. After reads with adapters were removed by using fastp software (fastp v0.19.4), all non-redundant transcripts were mapped with *Medicago sativa* reference genome (https://figshare.com/articles/dataset/Medicago_sativa_genome_and_annotation_files/12623960 (accessed on 1 November 2023)). Novel genes were screened by using StringTie. FPKM (Fragments Per Kilobase Million) values were calculated in accordance with the gene length. Differentially expressed genes (DEGs) were identified between comparisons by using DESeq2. *p*-values, and log_2_ fold changes were set as criteria for obvious differential expression. In accordance with the hypergeometric test, with pathway-based hypergeometric distribution checking for Kyoto Encyclopedia of Genes and Genomes (KEGG) and Gene Ontology (GO) term-based profiles for GO, enrichment analysis was conducted. We thank Wuhan Metware Biotechnology Co., Ltd. (Wuhan, China) for assistance with sequencing.

### 4.3. Phytohormone Analysis

Endogenous auxin, cytokinin (CK), abscisic acid (ABA), jasmonate (Ja), salicylic acid (SA), gibberellin (Ga), ethylene (ETH), strigolactone (SL), and melatonin (MLT) were quantified by using LC–MS/MS. Firstly, the samples (15 mg) were dissolved in 1 mL of methanol/water/formic acid (15:4:1, *v*/*v*/*v*) and frozen in liquid nitrogen. Then, 10 mL of the internal standard mixed solution (100 ng/mL) was added to the extract as internal standards (IS) for further quantifying. After the liquid was vortexed for 10 min and centrifugated (12,000 r/min, 5 min, and 4 °C), the supernatant was poured into microtubes. Subsequently, the supernatant was evaporated, dissolved in 100 μL 80% methanol, and filtered for further analyses. The UPLC and ESI-MS/MS conditions were introduced by Niu et al. [71]. The detected metabolites were annotated based on the KEGG compound database (http://www.kegg.jp/kegg/compound/ (accessed on 10 November 2023)).

### 4.4. qRT-PCR Analysis

We extracted the total RNA by using the RNA pure plant kit (Tb Green^®^ Premix Ex Taq™ II (TAKARA, Beijing, China)). Then, we obtained the first strand of the reverse-transcribed cDNA by using the specifications of the Monad first-strand cDNA Synthesis Kit. The primers were designed by utilizing PRIMER-BLAST (Appendix A). The ABI7500 quantitative PCR instrument was adopted to perform real-time fluorescence quantitative PCR. The gene (MsG0180001288.01) was selected as an actin gene for high and stable levels of expression in nine samples according to the FPKM value, and each group was identified from three repetitions.

## 5. Conclusions

In this work, we elucidated the molecular mechanism of floral bud development in alfalfa based on the phenotypic, metabolome, and transcriptome in three stages (D1, D2, D3). The transcriptome results revealed that the phytohormone biosynthesis and signaling pathway were closely associated with floral bud growth. The metabolomic results indicated that GA, IAA, CK, and JA were the critical phytohormones involved in floral bud differentiation and development. Notably, many key genes participating in phytohormone biosynthesis and signaling pathways might play a crucial role in regulating floral bud growth. Finally, we uncovered that many TF family members were closely correlated with floral bud development, such as *AP2*/*ERF*-*ERF*, *MYB*, *MADS*-*M*-*type*, *bHLH*, *NAC*, and *WRKY*. This work established regulatory networks related to phytohormones regulating floral bud development, providing potential leads for the molecular breeding of alfalfa.

## Figures and Tables

**Figure 1 plants-13-01078-f001:**
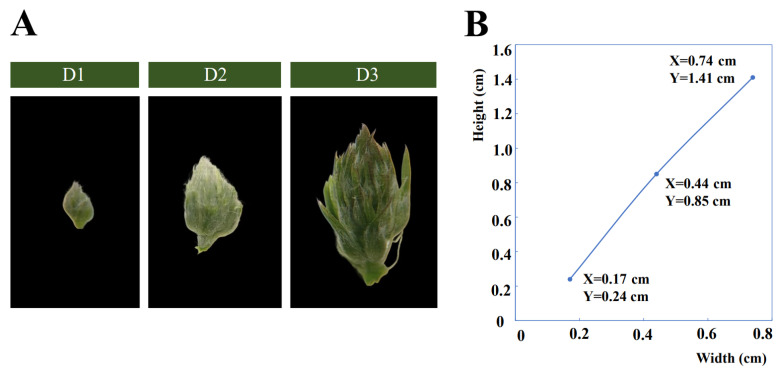
(**A**) The phenotypes of the three developmental stages (D1, D2, and D3) of floral buds in alfalfa. (**B**) Width and height of D1, D2, and D3; X and Y represent values of width and height of floral bud, respectively.

**Figure 2 plants-13-01078-f002:**
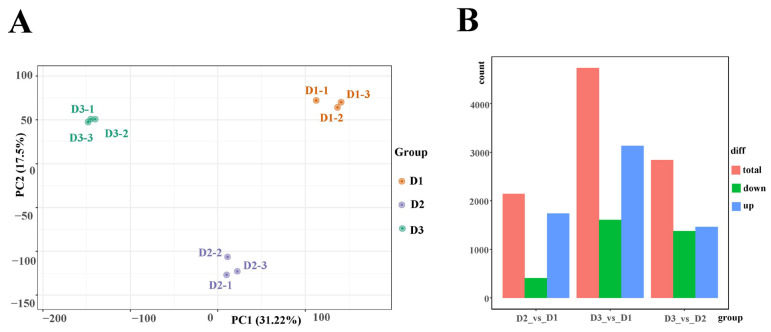
(**A**) PCA plot analysis of transcriptome; the *x*-axis and *y*-axis represent principal component 1 (PC1) and principal component 2 (PC2), respectively. (**B**) Statistics of differentially expressed genes in the D2 vs. D1, D3 vs. D1, and D3 vs. D2 comparison.

**Figure 3 plants-13-01078-f003:**
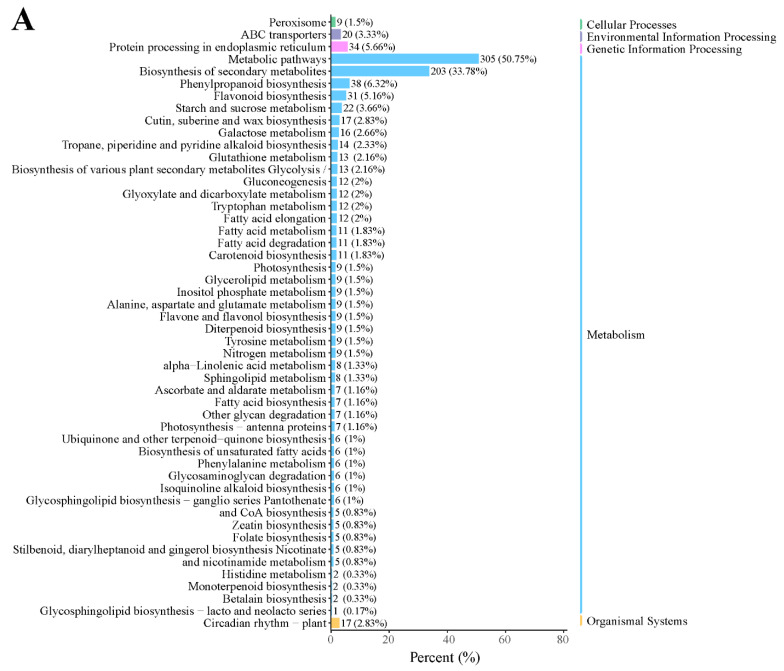
KEGG classification in the (**A**) D2 vs. D1 and (**B**) D3 vs. D2 comparison.

**Figure 4 plants-13-01078-f004:**
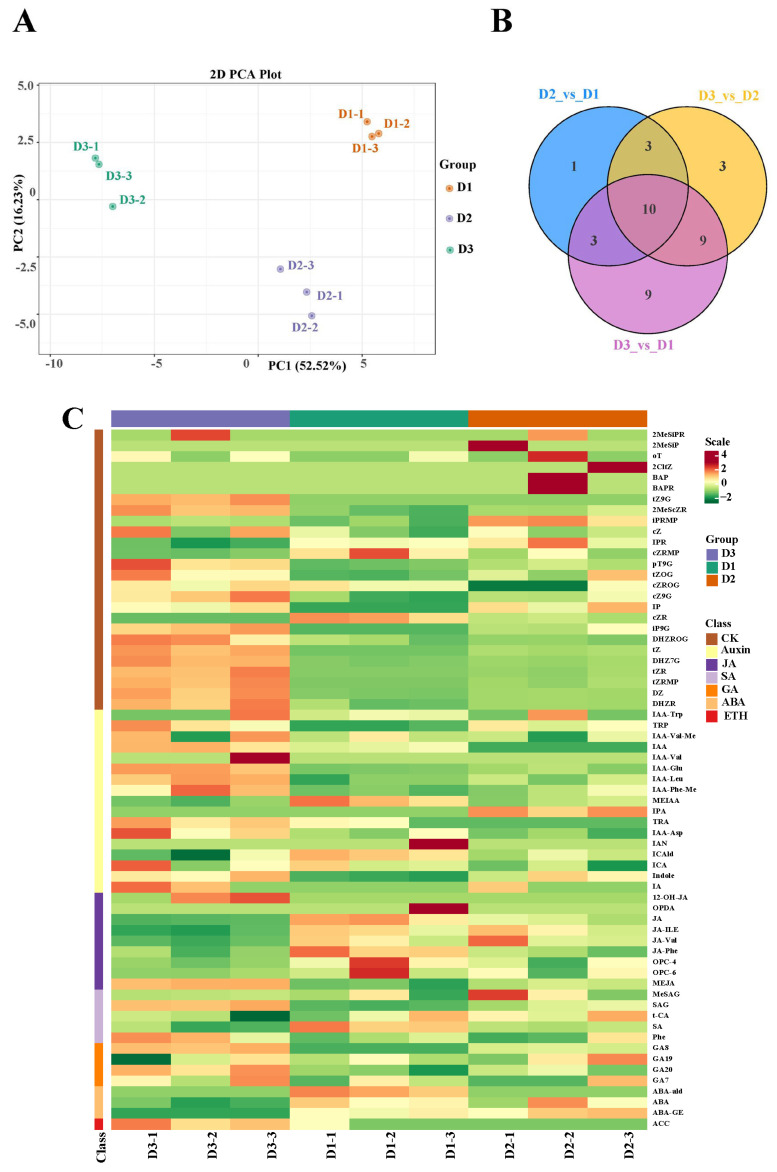
(**A**) PCA plot analysis of metabolome; the *x*-axis and *y*-axis represent principal component 1 (PC1) and principal component 2 (PC2), respectively. (**B**) Venn map of the three comparisons. (**C**) Differentially accumulated phytohormones in the three groups.

**Figure 5 plants-13-01078-f005:**
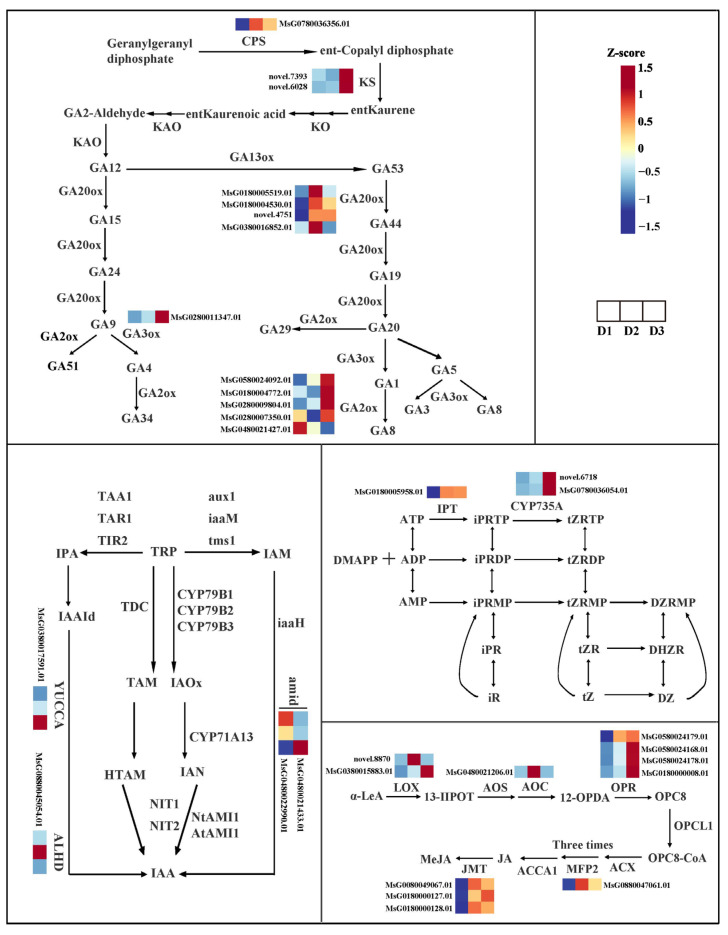
Gene expression of GA, IAA, CK, and JA biosynthesis pathways.

**Figure 6 plants-13-01078-f006:**
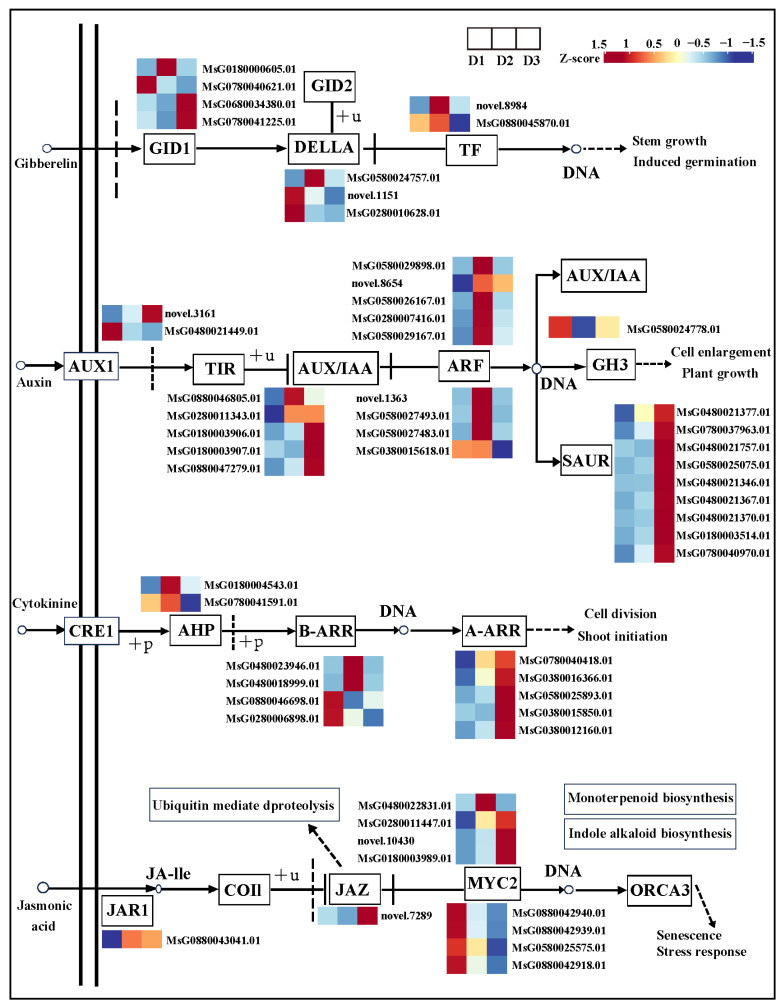
Gene expression of GA, IAA, CK, and JA signaling pathways.

**Figure 7 plants-13-01078-f007:**
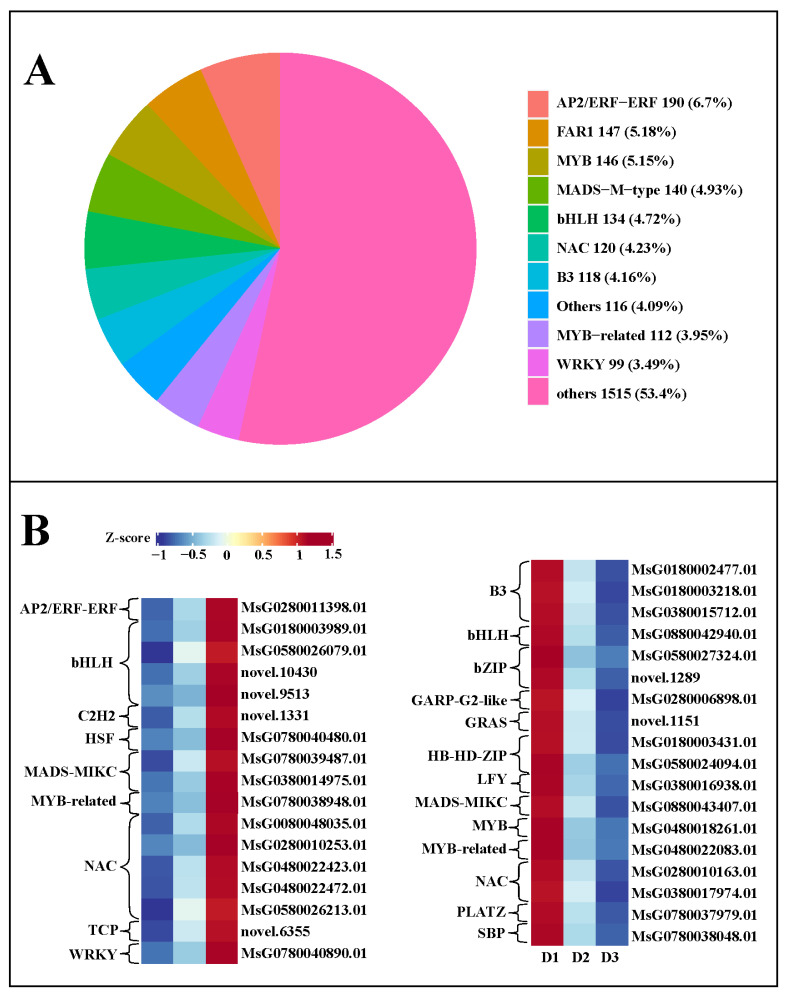
(**A**) Transcription factors in the three groups. (**B**) TFs with continuously increased and decreased expression levels in the developmental process of floral buds.

**Table 1 plants-13-01078-t001:** Differentially accumulated phytohormones in the D2 vs. D1 comparison.

Index	Compounds	Class	D2	D1	Log_2_FC
ABA-ald	Abscisic aldehyde	ABA	0	131.35	−Inf
IAA	Indole-3-acetic acid	Auxin	0	19.92	−Inf
IPA	3-Indolepropionic acid	Auxin	2.19	0	Inf
TRA	Tryptamine	Auxin	0	1.12	−Inf
iPRMP	N-6-iso-pentenyladenosine-5′-monophosphate	CK	10.14	4.77	1.09
pT9G	4-[[(9-beta-D-Glucopyranosyl-9H-purin-6-yl) amino] methyl] phenol	CK	0.96	0.46	1.06
cZROG	cis-Zeatin-O-glucoside riboside	CK	0.32	0.99	−1.64
cZ9G	cis-Zeatin-9-glucoside	CK	0.3	0.12	1.3
IP	N6-isopentenyladenine	CK	2.49	0.77	1.7
cZR	cis-Zeatin riboside	CK	1.07	3.62	−1.75
iP9G	N6-Isopentenyl-adenine-9-glucoside	CK	3	0	Inf
tZ	trans-Zeatin	CK	0.57	0	Inf
DHZ7G	Dihydrozeatin-7-glucoside	CK	0.21	0.09	1.13
tZR	trans-Zeatin riboside	CK	1.26	0	Inf
tZRMP	9-Ribosyl-trans-zeatin 5′-monophosphate	CK	3.17	0	Inf
GA8	Gibberellin A8	GA	13	0	Inf
MEJA	Methyl jasmonate	JA	99.53	43.5	1.19

**Table 2 plants-13-01078-t002:** Differentially accumulated phytohormones in the D3 vs. D2 comparison.

Index	Compounds	Class	D3	D2	Log_2_FC
ABA-GE	ABA-glucosyl ester	ABA	0.00	1755.20	−Inf
IAA	Indole-3-acetic acid	Auxin	34.59	0.00	Inf
IAA-Glu	Indole-3-acetyl glutamic acid	Auxin	22.23	5.64	1.98
IPA	3-Indolepropionic acid	Auxin	0.00	2.19	−Inf
TRA	Tryptamine	Auxin	2.39	0.00	Inf
IAA-Asp	Indole-3-acetyl-L-aspartic acid	Auxin	78.57	38.04	1.05
IA	3-Indoleacrylic acid	Auxin	10.98	4.37	1.33
tZ9G	trans-Zeatin-9-glucoside	CK	5.80	0.00	Inf
IPR	N6-isopentenyladenosine	CK	1.57	3.75	−1.26
pT9G	4-[[(9-beta-D-Glucopyranosyl-9H-purin-6-yl) amino] methyl] phenol	CK	2.25	0.96	1.23
cZROG	cis-Zeatin-O-glucoside riboside	CK	1.08	0.32	1.77
cZR	cis-Zeatin riboside	CK	0.00	1.07	−Inf
iP9G	N6-Isopentenyl-adenine-9-glucoside	CK	6.91	3.00	1.20
tZ	trans-Zeatin	CK	3.00	0.57	2.40
DHZ7G	Dihydrozeatin-7-glucoside	CK	1.25	0.21	2.59
tZR	trans-Zeatin riboside	CK	17.15	1.26	3.77
tZRMP	9-Ribosyl-trans-zeatin 5′-monophosphate	CK	27.54	3.17	3.12
DZ	Dihydrozeatin	CK	1.06	0.29	1.88
DHZR	Dihydrozeatin ribonucleoside	CK	2.27	1.12	1.02
ACC	1-Aminocyclopropanecarboxylic acid	ETH	30.17	0.00	Inf
GA8	Gibberellin A8	GA	27.36	13.00	1.07
GA7	Gibberellin A7	GA	0.43	0.20	1.13
12-OH-JA	12-Hydroxyjasmonic acid	JA	249.96	0.00	Inf
MEJA	Methyl jasmonate	JA	203.97	99.53	1.04
t-CA	trans-Cinnamic acid	SA	47.27	109.61	−1.21

## Data Availability

All data are open and available. The raw data are available in the NCBI (BioProject ID PRJNA1049519) (https://www.ncbi.nlm.nih.gov/sra/ (accessed on 6 December 2023)).

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
