# Peer review of "Integrative Metabolomic and Transcriptomic Analysis Elucidates That the Mechanism of Phytohormones Regulates Floral Bud Development in Alfalfa"

_plants, 2024, doi:10.3390/plants13081078_

Round 1

Reviewer 1 Report

Comments and Suggestions for Authors

This manuscript analyzed the expression of phytohormones involved in the process of alfalfa floral bud differentiation in great detail at the transcriptome level and metabolomic level and presented the results. There are a few minor corrections, which are noted directly in the attached manuscript.

Reviewer 2 Report

Comments and Suggestions for Authors

The current paper devoted to investigation the mechanism of floral bud induction and development in alfalfa. Authors performed analysis of transcriptome and metabolome analysis at several stages of buds development and provide data about some gene expression at mRNA level and metabolome presence on it.

From the scientific point of view, it is not clear why authors choose these developmental stage. Moreover, the samples for analysis contains different cell type and gradients of hormones, differences in gene expression and different epigenetic status. The main question: did differences shown by authors represent kinetics of development or differences in cell type/status.

The most important ist he confirmation of your data with in situ investigation: localization of gene expression in certain cell type, hormone abundance in specifci routes etc. Authors may know that, for example, polar distribution of auxin is a key for creating flowers arhitecture. Total measurement can not provide you such information.

 This point need to be carefully discussed.

Moreover, what did authors mean as differentiation and development? I would suggest to keep only development or growth and development.

The most interesting stage in floral bud development is a very early stage (intiation), but this stage is missing in the paper. 

There are also other points.

In the introduction authors in each sentence write word „alfalfa“ what is redundant: you study only alfalfa!

Line 18: which YUCCA do you mean? There are up 10 YUCCA genes in Arabidopsis and tomato and each have own function, expresed under different developmental stages. Wahy amid is in italic?

Lines 11, 21, 35 etc – grammar. Please, check carefully for punctuations, clarity etc.

Figure 1: no scale bar. Images not enough informative.

Line 186: “GA, IAA, CK, and JA were the critical phytohormones“ – this is correct, but the same is true for all plant organs. Most importnat is place of synthesis and cell type of action!

Line 203: what is AUCCA? Maybe YUCCA? Which one? Where it localise? Flower bud can be consider as auxin routes originated from YUCCA biosynthesis site. More information about localization of hormone synthesis and gradients are required, at least for discussion.

Figures 5 and 6: nice models, but you need to specify cell type. At the stage D3 you have cells with extremely different hormone response and this differences made flowers development.

Line 358: what is alfalfa material? Which conditions have you used?

Comments on the Quality of English Language

Many correction s for clarity are required see details in the comments.

Round 2

Reviewer 2 Report

Comments and Suggestions for Authors

Thank you for the very detailed answer.

The text is much better, but still some points need to be clarified.

Stage of development is very good for study anatomy, but not fit well for study mechanism. This stage allow you to see results of „molecular action“, but not action in progress.

Please, consider it for the future.

For the hormone measurement: we can tell rather about balance between hormone synthesis, action, degradation, transport and conjugation.

Specifically, in the case of auxin it concentration may have several opposite meaning. Auxin act in specific cells and total concentration is not so important. Flowers is auxin gradient and auxin canalisation.  You need to mention this in discussion.

For the detailed investigation you can easy fix flowers at the all time points, cleafing, made immunolabeling and scan in 3D. For the simple variant, you can made a section (10- 20 µm) with microtom and observed all phenptype and study gene expression.
